# Immune and Oxidative Stress Response of the Fish *Xyrichthys novacula* Infected with the Trematode Ectoparasite *Scaphanocephalus* sp. in the Balearic Islands

Amanda Cohen-Sánchez [1], Antoni Gabriel Sánchez-Mairata [1], José María Valencia [2,3], Antonio Box [4], Samuel Pinya [5,6], Silvia Tejada [6,7,8] and Antoni Sureda [1,6,8,*]

[1] Research Group in Community Nutrition and Oxidative Stress, University of Balearic Islands, 07122 Palma de Mallorca, Spain; amandacohen.tic@gmail.com (A.C.-S.); tonigabriel2000@gmail.com (A.G.S.-M.)
[2] Laboratorio de Investigaciones Marinas y Acuicultura (LIMIA)-Govern de les Illes Balears, Instituto de Investigación y Formación Agroalimentaria y Pesquera de las Illes Balears (IRFAP), 07157 Port d'Andratx, Spain; jmvalencia@dgpesca.caib.es
[3] Instituto de Investigaciones Agroambientales y de Economía del Agua (INAGEA) (INIA-CAIB-UIB), 07122 Palma de Mallorca, Spain
[4] Department of Agricultura, Ramaderia, Pesca, Caça i Cooperació Municipal, Consell Insular d'Eivissa, 07800 Eivissa, Spain; tonibox@conselldeivissa.es
[5] Interdisciplinary Ecology Group, Department of Biology, University of the Balearic Islands, 07122 Palma de Mallorca, Spain; s.pinya@uib.es
[6] Health Research Institute of Balearic Islands (IdISBa), 07120 Palma de Mallorca, Spain; silvia.tejada@uib.es
[7] Laboratory of Neurophysiology, Biology Department, University of the Balearic Islands, 07122 Palma de Mallorca, Spain
[8] CIBER Fisiopatología de la Obesidad y Nutrición (CIBEROBN), Instituto de Salud Carlos III (ISCIII), 28029 Madrid, Spain
* Correspondence: antoni.sureda@uib.es; Tel.: +34-971172820

**Abstract:** Global change produces substantial modification to the distribution and rhythm of infection of diseases in fish, favouring the introduction of new pathogens. Recently, the presence of black spot disease, associated with a digenean fluke of the genus *Scaphanocephalus*, has been observed in specimens of *Xyrichthys novacula* on the island of Ibiza (Balearic Islands). The aim of the present study was to evaluate the antioxidant and immune response in both the skin mucus and spleen of *X. novacula* depending on the degree of infection by *Scaphanocephalus* sp. The specimens were captured in a control area, without the presence of the parasite, and in an affected area, classifying the fish as low infection (1–15 spots) and high infection (>15 spots). As the degree of infection increased, a decrease in the body condition index was observed. The activity of the antioxidant enzymes in the mucus—catalase, superoxide dismutase and glutathione peroxidase—increased progressively with the degree of infection. This activation of antioxidant defences was not enough to prevent an increase in malondialdehyde levels, an indicator of oxidative damage, in the group with the highest infection. Similarly, an increase in immunological parameters—lysozyme, alkaline phosphatase, myeloperoxidase and immunoglobulins—was observed in mucus as infection increased. Regarding the spleen, only an increase in lysozyme activity and alkaline phosphatase in fish with a greater severity of infection was observed. In conclusion, as the severity of *Scaphanocephalus* sp. infection increased, it induced an immune and oxidative stress response in skin mucus, leading to a decrease in overall body condition. The potential health effects that the ectoparasite may have on *X. novacula* populations will require follow-up studies.

**Keywords:** oxidative stress; immune system; Balearic Islands; pearly razorfish; ectoparasite

**Key Contribution:** The pearly razorfish, *Xyrichthys novacula*, has recently been affected by a fluke ectoparasite of the genus *Scaphanocephalus* in waters of Ibiza Island (Spain). The parasite induced a progressive antioxidant and immune response in the mucus of *X. novacula*, whereas in the spleen, the response was only observed in highly affected specimens.

## 1. Introduction

Global changes encompass various large scale environmental alterations and transformations, which result from anthropogenic activities. In the last fifty years, humans have altered the structure and functioning of ecosystems around the world, faster and in a larger number of ways than ever in the history of humanity, producing a substantial and mostly irreversible loss of life's diversity. In this sense, the introduction of alien species due to human activity has been one of the most important processes for species loss [1]. Moreover, globalisation, and the increase in commercial trade, is one of the main causes of increase in exotic species, which causes a problem for native species in an area, particularly if the new species is invasive in character [2,3]. The most common introduction source in the Mediterranean is directly associated with vessels, as species can adhere to their surface or enter in ballast waters, although they can also be transmitted by the movement of animals, mainly birds and fish [4,5]. These species can lead to a displacement of native species and can change trophic webs in the ecosystem [6].

In this sense, global change can produce substantial modifications to the distribution and rhythm of infection of diseases in animals, such as the introduction of new pathogens in populations with little immunologic resistance [7]. Thus, the increase in temperature can be a grave danger to the host–parasite relationship in aquatic environments, as it can lead to an increased risk of new vectors and new diseases in the ecosystem [8]. Moreover, an increase in temperature could imply higher reproduction taxa for parasites, which could accelerate transmission and augment its abundance. In addition, human pollutants, in particular organic persistent pollutants, can have a strong impact on the host–parasite relationship [9]. As a matter of fact, whilst many parasites are extremely sensitive to change, others are more resistant than their hosts and can even thrive in polluted environments. Generally, endoparasite infections with complex indirect cycles tend to decrease whilst ectoparasites increase with pollution [10]. This fact is probably due to the constant exposure of ectoparasites to the environment which have evolved to be more flexible and resistant to environmental changes [11].

Black spot disease is one of the most common signs of ectoparasite infection [12]. More than 30 species of ectoparasites have been found to cause blackspot disease, one of which is the genus *Scaphanocephalus*, a group of parasitic trematodes of the subclass digenea. Digenean trematodes have a complex life cycle that involves several hosts and developmental stages. The parasite's life cycle requires coastal molluscs as the primary host, marine fish as the secondary host and fish-predating birds, such as *Pandion haliaetus*, as definitive hosts, although this has not been conclusively established [13,14]. In this sense, since in the final host *Scaphanocephalus* acts as an endoparasite, in fish where it produces pigmented dermatopathies, presented as focal circular spots or papules, it would act more as an ectoparasite or cutaneous endoparasite. Black spot disease caused by *Scaphanocephalus* has been found in invertebrates such as *Penaeus vannamei* (Boone, 1931) [15], marine fish such as *Bolbometopon muricatum* (Valenciennes, 1840), one of the largest parrot fish [16], and even in freshwater fish such as *Esox lucius* (Linnaeus, 1758) [17]. Up to now, only three species of this genus have been described, including *S. australis* (Johnston, 1917), *S. adamsi* (Iwata, 1997) and *S. expansus* (Creplin, 1842).

The presence of parasites can cause physiological and behavioural alterations in hosts, favouring new infections or their predation, and endangering their survival [18]. To cope with ectoparasites, fish have developed an external mucosal layer that acts as both a physical and biochemical barrier. Among the components of mucus are immune

elements, such as immunoglobulins and lysozyme, and antioxidant enzymes to reduce oxidative damage derived from the infectious process [19,20]. In this sense, immunological parameters and markers of the oxidative state in mucus can provide information on the evolution of the infection and the host's response capacity. Also, the spleen as a lymphoid organ in fish involved in immune reactivity and blood cell formation could be affected by parasitic infections [21]. In fact, it has also been suggested that splenic responses can reflect adaptation to the level of parasite load in the host [22].

*Xyrichthys novacula* (Linnaeus, 1758), or pearly razorfish, is a wrasse of the family Labridae, which can grow up to 20 cm in length [23]. This species is of great importance in the Balearic Islands, both for human consumption and recreational fishing [24,25]. This popularity has led to its overfishing, which resulted in an administrative ban that prohibits the fishing of this species from April to September to protect the mating season. During the fishing season of 2015, some fishermen reported white spots on some of the captured individuals in the south of Ibiza Island; the cause was later identified by the presence of *Scaphanocephalus* sp. In a previous study, a immune response in the mucus and the induction of antioxidant mechanisms in the mucus and liver, as well as metabolic activation, was observed in the presence of the parasite in *X. novacula* [26]. In the year 2022, a very significant increase in the number of spots on fish was observed (researchers' personal observations), with specimens exceeding 100 spots on their surface. Due to this notable increase in the presence of *Scaphanocephalus* compared to 2020 [26], the aim of this study was to analyse the immune and antioxidant response in the mucus and the spleen of *X. novacula* depending on the degree of infection.

## 2. Materials and Methods

### 2.1. Sample Recollection

The sampling areas were located in the Pityusic Islands, Spain, the name given to the group formed by the islands of Ibiza and Formentera, and the islets surrounding them. The two fishing spots were the sandy bottoms of Es Cubells in Ibiza and Sa Mola in Formentera. Es Cubells, located in the southern part of the island of Ibiza, is the area where specimens of *X. novacula* with the presence of spots were observed for the first time and represents the affected area, while Sa Mola, located in the southeast of the island of Formentera, is considered the control area with an absence of fish affected by the spots (Figure 1). From Es Cubells, exploratory samplings were carried out to determine how far away infected fish were caught in order to establish a control area without evidence of the parasite. The control area was selected to reduce the possibility that the fish had been in contact with the parasite, even if they showed no evidence of infection.

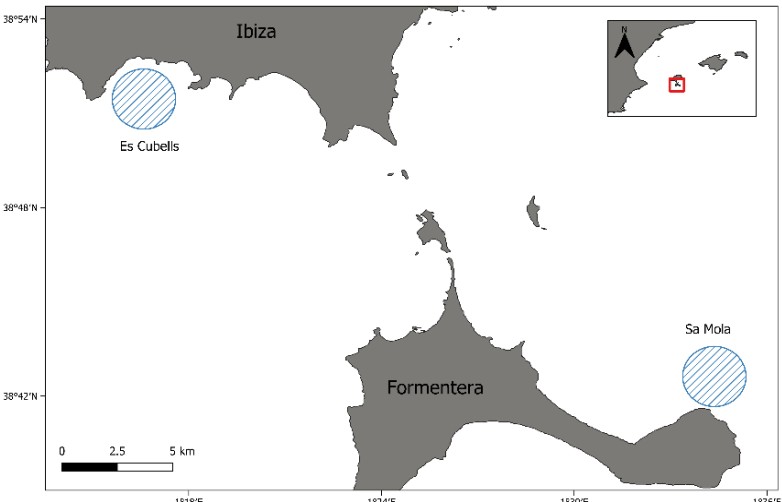

**Figure 1.** Sampling sites in Es Cubells (Ibiza) where the specimens of *Xyrichthys novacula* infected by *Scaphanocephalus* sp. can be found, and Sa Mola (Formentera), considered the control site.

The specimens were sampled during the first week of October 2022 to avoid the fishing ban period, established to protect the reproductive season of the fish and to reduce variability in climatic conditions. Fish were caught by hook and line with worms as bait at a similar depth of 18–20 m. In the Sa Mola area, a total of 10 female specimens were captured without the presence of ectoparasites. A total of 77 fish were caught in the Es Cubells area. Of these fish, those that did not have parasites (10 specimens) and also the males (6 specimens), due to the low number of captured, were discarded for further analysis. Also, in order to use fish of the same size and reduce bias due to size and potential exposure time to the parasite, since *X. novacula* are protogynous sequential hermaphrodites, it was decided to work only with females. Additionally, the intestinal cavities and the stomach/gut of all initially selected specimens were inspected to detect the possible presence of endoparasites, as well as the gills that could interfere with the study. Endoparasites were observed in the intestinal cavity of three fish from Es Cubells and were directly discarded from the study, so three new specimens free of parasites were incorporated. Finally, 20 specimens with the presence of spots with a size similar to that of the control zone were selected, and the rest were returned to the sea with minimum manipulation.

The fish were immediately anesthetised with tricaine methane sulfonate (MS-222) (1 g in 10 L of marine water) [27], the spleens were dissected and mucus samples (about 0.75 mL) from the skin were carefully collected with a spatula, and care was taken to avoid the scales. Due to the small size of the fish, mucus was obtained from the dorsal area of both sides of the specimens. The fish were weighed and measured, and the abundance of epithelial spots, due to the presence of the parasite, was determined visually. The body condition index was determined using the Fulton's condition factor (Kc) with the formula $Kc = 100 \times (weight (g)/length (cm)^3)$. The fish from Es Cubells were divided into two groups of 10 according to the value of the median of the spots of the infected fish: (1) low infection, between 1–15 parasites and (2) high infection with more than 15 parasites (Figure 2).

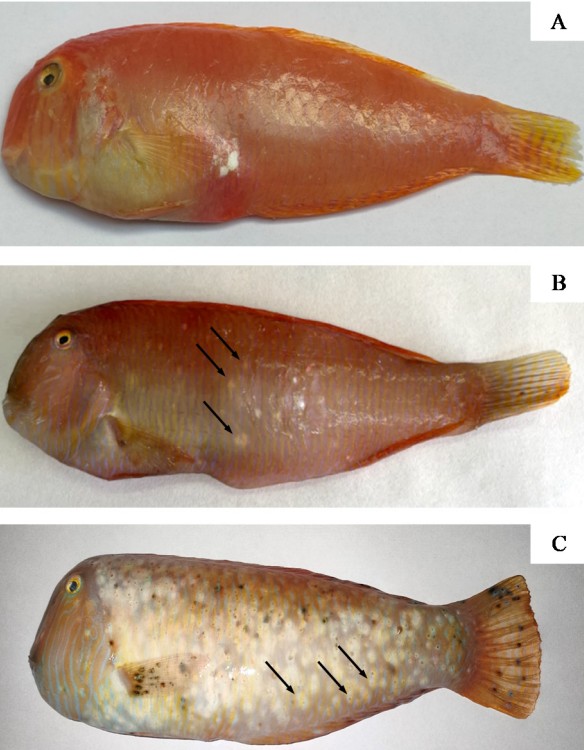

**Figure 2.** Representative images of *Xyrichtys novacula* without evidence of infection (**A**), low infection (**B**) and high infection (**C**) where black spots due to melanin accumulation can be observed. Black arrows show some of the spots.

Once obtained, the mucus samples were placed into liquid nitrogen where they were kept until reaching the laboratory where they were stored at −80 °C until later use. For excystation of metacercariae from fish skin preserved in ethanol, two syringe needles (30G × 1/2″) were used and visualised under an Olympus SZH stereomicroscope (Olympus Optical Co., Ltd., Tokyo, Japan) with a HD camera at LIMIA facilities (Figure 3). The experimental procedure followed was based on EU Directive 2010/63/UE for animal experimentation and was reviewed and approved by the Ethics Committee for Animal Experimentation of the University of the Balearic Islands (Reference 020/06/AEXP).

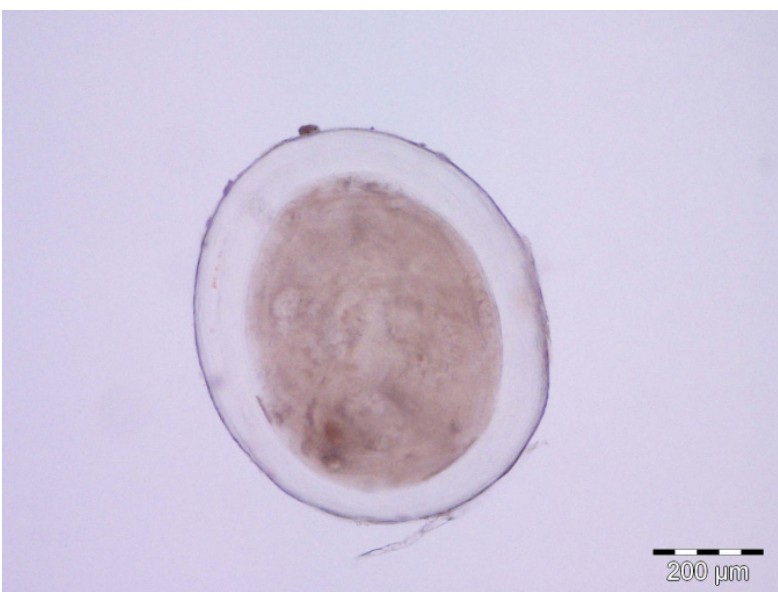

**Figure 3.** Representative image of *Scaphanocephalus* sp. metacercaria removed from *Xyrichtys novacula*.

### 2.2. Molecular Analysis

The genus of the ectoparasite was confirmed as *Scaphanocephalus* using the previously described methodology based on 28S rDNA amplification [26]. Briefly, a total of 14 metacercaries from different specimens were dissected from epithelial sections containing skin papules using dissecting needles. To confirm whether the different types of spots (black and white) were caused by the same parasite, 8 spots of each type were analysed. DNA extraction was then carried out using a tissue DNA extraction kit (NucleoSpin Tissue XS extraction kit, Macherey-Nagel, Dueren, Germany) following the instructions provided by the manufacturer. Once the DNA was purified, the 28S rDNA was partially amplified, including the variable expansion regions D1-D3 of the parasites using the polymerase chain reaction (PCR). The used primers were 1500R (5′-GCT ATC CTG AGG GAA ACT TCG-3′) and LSU5 (5′-TAG GTC GAC CCG CTG AAY TTA AGC-3′) [28,29]. PCR reactions were performed using the KAPA Taq Ready Mix DNA Polymerase (KapaBiosystems Inc., Corston, Bath, UK) with the following conditions: a denaturation period of 5 min at 95 °C followed by 40 cycles of 95 °C for 30 s, 57 °C for 30 s, 72 °C for 1 min and a final stage of 72 °C for 10 min.

PCR products were separated in 1% agarose and stained with GelRed® Nucleic Acid Gel Stain (Biotium, Fremont, CA, USA). Then, the obtained amplicons were purified with the mi-PCR purification kit (Metabion International, Planegg, Germany) and sequenced on a 3130xl automated DNA sequencer (Applied Biosystems, Waltham, MA, USA). Sequences were finally aligned and edited using the BioEdit 7.1.3.0 software package [30] and compared using the National Centre for Biotechnology Information (NCBI) Basic Local Alignment Search Tool (BLAST+ v2.13). Phylogenetic and molecular evolutionary analysis was conducted using MEGA version 6.

## 2.3. Biochemical Analysis

Prior to the biochemical analysis, the mucus samples were centrifuged at $1500\times g$ for 10 min at 4 °C and the supernatants were recovered. The spleens were homogenised in 100 mM Tris–HCl buffer, pH 7.5, with Ultra-Turrax® Disperser (IKA-Werke, Staufen, Germany) and centrifuged at $9000\times g$, for 10 min, 4 °C (Sigma 3K30, Osterode am Harz, Germany). After centrifugation, supernatants were collected and used for the biochemical analysis. The activities of the antioxidant enzymes, MPO and ALP, and the concentration of protein carbonyls were determined in a Shimadzu UV-2401 spectrophotometer (Shimadzu Corporation, Kyoto, Japan) at 25 °C. The activities of CAT, SOD and GPX were determined following previously described methodologies [31–33]. MPO activity was monitored by analysing the oxidation of 2-methoxyphenol [34]. ALP activity in mucus was determined at 405 nm using p-nitrophenyl phosphate as a substrate [35]. Lysozyme activity was determined at 450 nm in a microplate reader (BioTek®, PowerWaveXS, Agilent Technologies, Madrid, Spain) using a bacterial suspension of *Micrococcus lysodeikticus* [36]. Total immunoglobulin (Ig) concentration was quantified after precipitation with 12% 10,000 kDa polyethylene glycol for 2 h [37]. MDA concentration in mucus was determined using a colorimetric assay kit (Merk Life Science S.L.U., Madrid, Spain). Protein carbonyls were assayed in mucus with the method of Levine using 2,4-dinitrophenylhydrazine (DNPH) as a chromogenic agent and measured at 360 nm [38]. Total protein content was determined in a microplate reader (BioTek®, PowerWaveXS) with a colorimetric method (Merk Life Science S.L.U., Madrid, Spain), using bovine serum albumin (BSA) as a standard to normalise biochemical results. All biochemical analyses were performed in duplicate.

## 2.4. Statistical Analysis

The results were analysed with the statistical package (SPSS 27.0 for Windows®, IBM® SPSS Inc., Chicago, IL, USA). Firstly, the analysis of the normality of the data was carried out by applying the Shapiro–Wilk test and the homogeneity of variance with the Levene test. Statistical significance was then determined for all biomarkers by one-way analysis of variance (ANOVA). For those data groups that showed statistical differences, a post-hoc analysis was performed by applying the Bonferroni test, to determine the groups involved in the differences. Bivariate correlations between the number of spots and the Fulton's condition factor, size and biochemical analysis were calculated with Pearson's test. Values are shown as mean $\pm$ standard error and significance level was set at $p < 0.05$.

## 3. Results

### 3.1. Fish and Parasite Characteristics

The captured fish had an average length of $14.8 \pm 0.4$ cm (10.1–20.70 cm) and a weight of $48.0 \pm 3.3$ g (14–88 g). No differences in size were observed between the fish depending on the group ($14.8 \pm 0.6$ cm without infection; $14.9 \pm 0.8$ cm low infection; $15.0 \pm 0.9$ cm high infection, $p = 0.972$). The prevalence of affected fish with *Scaphanocephalus* sp. in the Es Cubells area was 87.0%, while in Sa Mola area no affected fish were observed. The fish from Es Cubells were classified into two groups according to the degree of infection: low infection group with 1–15 spots ($5.6 \pm 0.9$ spots) and high infection group with more than 15 spots ($49.9 \pm 11.8$ spots). The global mean number of spots on the skin of *X. novacula* specimens at Es Cubells was $27.6 \pm 6.1$. It should be noted that in one of the specimens analysed a total of 114 spots were observed. Moreover, the inside of the operculum was examined for black spots, with no evidence in any of the fish. The cysts are oval in shape with a fibrous capsule that surrounds it. On the fish, they appear as whitish spots in the early stages that change to black as the surrounding tissue starts to accumulate melanin.

The results of the Fulton's condition factor show a progressive decrease in its values between the fish without infection ($1.51 \pm 0.06$), low infection ($1.44 \pm 0.03$) and high infection ($1.32 \pm 0.05$), with statistically significant differences between the groups without infection and high infection ($p = 0.033$).

### 3.2. Molecular Analysis and Phylogenetic Analyses

Adequate amplification was obtained in 8 of the samples analysed (5 coming from white spots and 3 from black spots), resulting in a sequence with 1283 base pairs of the LSU rDNA gene after aligning the sequences of the different PCR amplicons. The sequences obtained showed an identity between 98.4 and 98.6%, with the 4 available sequences of *Scaphanocephalus* sp. in GenBank (MN160569, MN160569, MT461356, MK680936). GenBank accession numbers for the obtained sequences were OK045681 to OK045688.

A phylogenetic tree of 28rDNA (Figure 4) placed the obtained sequences in a clade with other *Scaphanocephalus* species (*S. expansus*, *Scaphanocephalus* sp. from *Acanthurus chirgus* and *Scaphanocephalus* sp. from *Siganus argenteus*), *Cryptocotyle* lingua and *Euryhelmis costaricensis*. All of them are digenean from the family Heterophyidae.

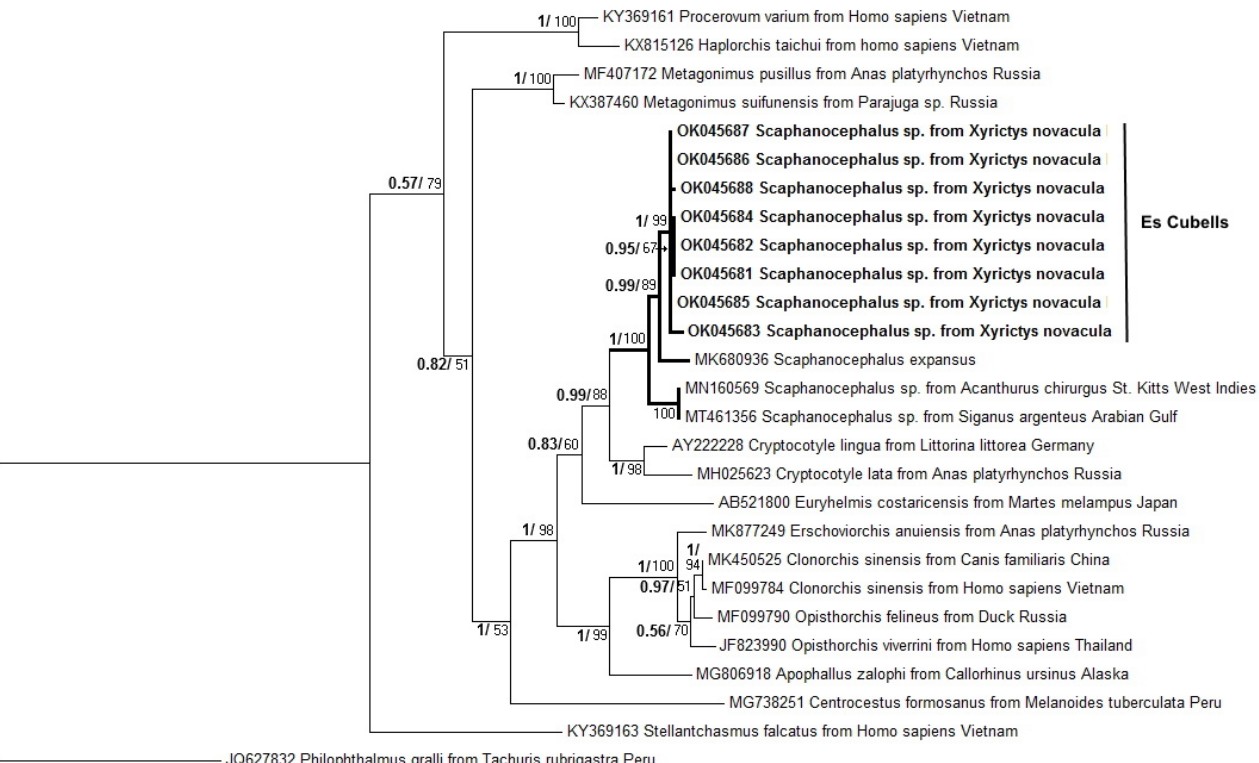

**Figure 4.** Maximum likelihood topology of digenean taxa based on 28rDNA partial sequences. Nodes show posterior probabilities (numbers in bold) and bootstrap support values. Novel sequences from this study are in bold and group with *Scaphanocephalus expansus*, *Scaphanocephalus* sp and *Cryptocotyle lingua*, heterophyid digeneans known to cause black spot disease in fishes in the Atlantic and Mediterranean. *Philophthalmus gralli* was used to root the tree. All GenBank accession numbers are given before the taxon branch labels.

### 3.3. Oxidative Stress Parameters

The activities of the antioxidant enzymes in the mucus of *X. novacula* are presented in Figure 5. The activities of all the enzymes—CAT, SOD and GPX—increased progressively with an increasing degree of infection. The activities of the three enzymes analysed were statistically higher in the group with high infection when compared to the control group ($p < 0.001$ for CAT; $p = 0.002$ for SOD; $p = 0.010$ for GPX). Additionally, CAT activity in the low infection group was also statistically higher than in the control group ($p = 0.013$).

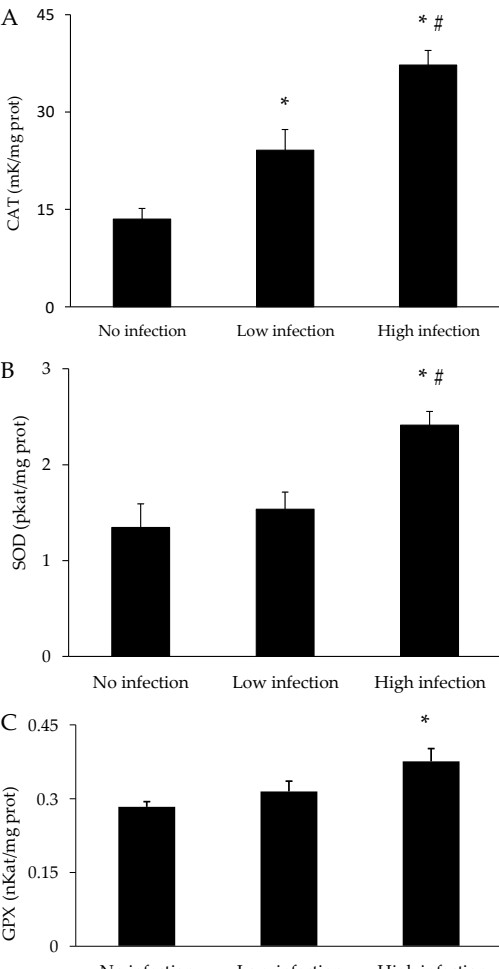

**Figure 5.** Activity of antioxidant enzymes—catalase (CAT) (**A**), superoxide dismutase (SOD) (**B**) and glutathione peroxidase (GPX) (**C**)—in mucus of *X. novacula*. Results are shown as mean ± S.E.M. (*) Significant differences with respect to the non-infected group; (#) Significant differences with respect to the low infection group (one-way ANOVA, $p < 0.05$).

The levels of MDA and protein carbonyls in the mucus of *X. novacula* are shown in Figure 6. The values of both parameters remained similar in the group with low infection when compared to the control, while they significantly increased in the group with high infection compared to the control ($p < 0.001$ for both parameters) and low infection groups ($p = 0.005$ for MDA and $p = 0.001$ for protein carbonyls).

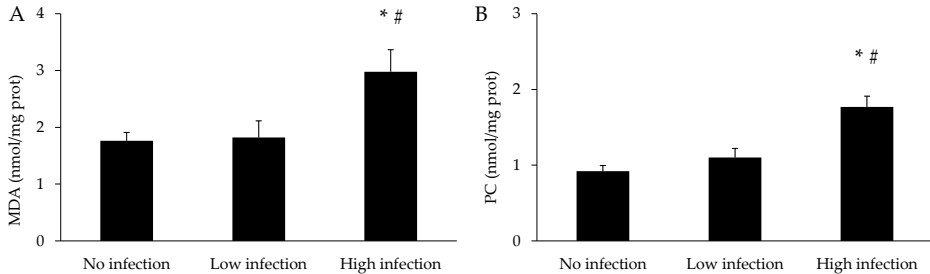

**Figure 6.** Concentration of malondialdehyde (MDA) (**A**) and protein carbonyls (PC) (**B**) in mucus of *X. novacula*. Results are shown as mean ± S.E.M. (*) Significant differences with respect to the no infected group; (#) Significant differences with respect to the low infection group (one-way ANOVA, $p < 0.05$).

### 3.4. Immune Parameters

The results of the immune parameters in the mucus of *X. novacula* are presented in Figure 7. Similar to antioxidant enzymes, the different immunological markers increased in parallel with the degree of infection. Lysozyme, ALP and MPO activity, and Ig levels in the mucus were significantly higher in the highly infected group when compared to the control ($p < 0.001$ for lysozyme; $p = 0.004$ for ALP; $p < 0.001$ for MPO; $p < 0.001$ for Ig) and the low infection groups ($p < 0.001$ for lysozyme; $p = 0.023$ for ALP; $p = 0.038$ for MPO; $p = 0.001$ for Ig). In addition, lysozyme and MPO activities are also higher in the low infection group when compared to the control ($p = 0.023$ for lysozyme; $p = 0.045$ for MPO).

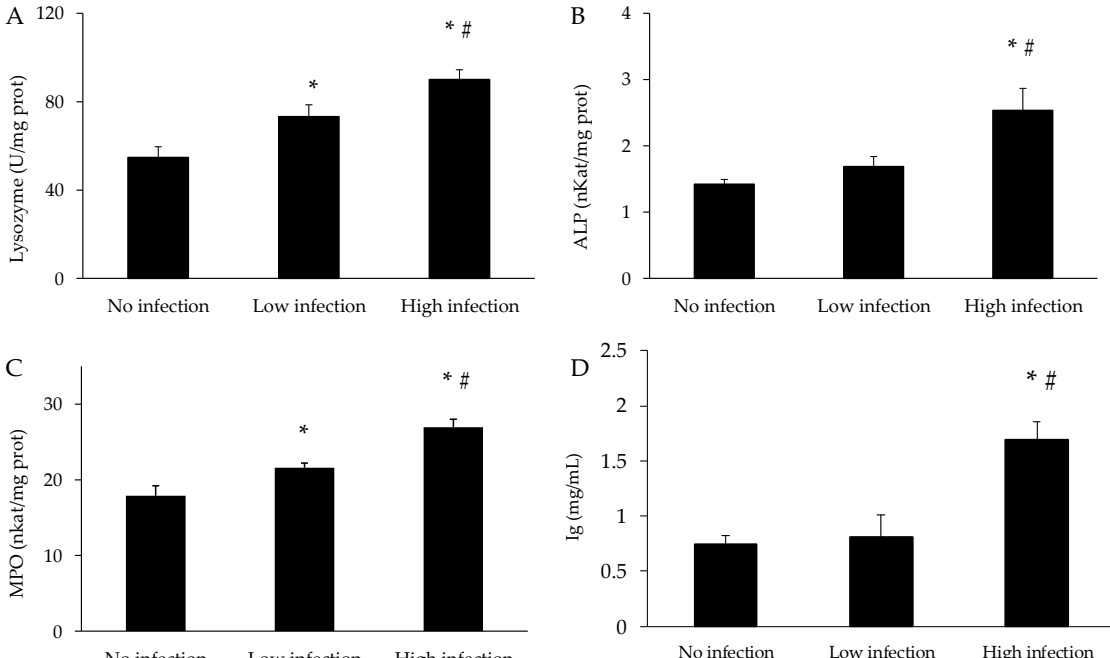

**Figure 7.** Activity of lysozyme (**A**), alkaline phosphatase (ALP) (**B**), myeloperoxidase (MPO) (**C**) and immunoglobulin levels (Ig) (**D**) in mucus of *X. novacula*. Results are shown as mean ± S.E.M. (*) Significant references with respect to the no infected group; (#) Significant differences with respect to the low infection group (one-way ANOVA, $p < 0.05$).

The activities of lysozyme and ALP in the spleen of *X. novacula* are shown in Figure 8. Both activities significantly increased in fish with high infection compared to the absence of infection ($p = 0.020$ for lysozyme and 0.036 for ALP) or low infection ($p = 0.018$ for lysozyme and 0.037 for ALP). No significant differences were found between non-infected fish and low infection.

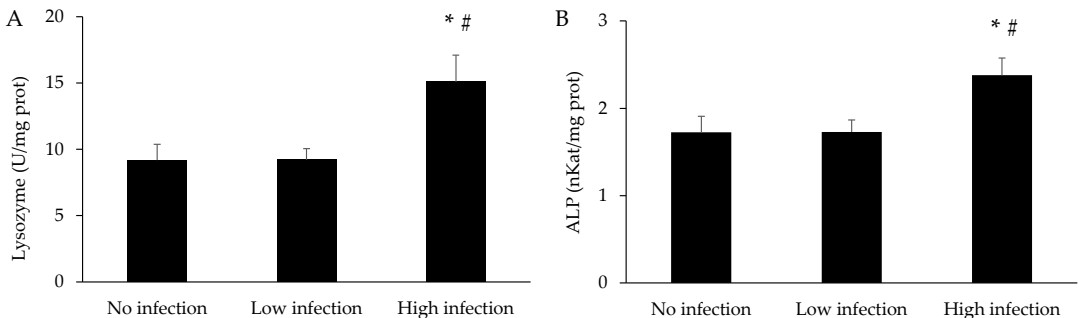

**Figure 8.** Activity of lysozyme (**A**) and alkaline phosphatase (ALP) (**B**) in the spleen of *X. novacula*. Results are shown as mean ± S.E.M. (*) Significant differences with respect to the non-infected group; (#) Significant differences with respect to the low infection group (one-way ANOVA, $p < 0.05$).

### 3.5. Bivariate Correlations

The analysis of correlations between the presence of spots and the different parameters analysed in mucus is presented in Table 1. An inverse correlation is observed between the number of spots and body condition, and a direct correlation with the rest of the oxidative stress and immunological parameters. No significant correlations were observed in the spleen between spots and lysozyme ($r = 0.170$; $p = 0.352$) or ALP ($r = 0.022$; $p = 0.910$). Moreover, no correlation was found between spots and the fish size ($r = 0.271$; $p = 0.156$).

**Table 1.** Bivariate correlations between the number of spots and the different parameters determined in the mucus of *X. novacula*.

|  |  | Kc | CAT | SOD | GPX | MDA | PC | LYZ | ALP | MPO | Ig |
|---|---|---|---|---|---|---|---|---|---|---|---|
| Spots | *r* | − 0.472 ** | 0.630 ** | 0.635 ** | 0.575 ** | 0.524 ** | 0.809 ** | 0.589 ** | 0.379 * | 0.684 ** | 0.529 ** |
|  | *p* | 0.010 | 0.000 | 0.000 | 0.001 | 0.004 | 0.000 | 0.001 | 0.043 | 0.000 | 0.003 |

Abbreviations: Kc, Fulton's condition factor; CAT, catalase; SOD, superoxide dismutase; GPX, glutathione peroxidase; MDA, malondialdehyde; PC, protein carbonyls; LYZ, lysozyme; ALP, alkaline phosphatase; MPO, myeloperoxidase; Ig, total immunoglobulins. Bivariate Correlations (*) Indicates a correlation at $p < 0.05$. (**) Indicates a correlation at $p < 0.01$.

## 4. Discussion

Globalisation together with climate change can influence the spatial and temporal distribution of pathogens that can find new environments conducive to their development, and the Mediterranean Sea is not an exception [39,40]. At surface level, marine heat wave indices have considerably increased over the last four decades, from 1982 to 2020, in the Balearic Islands, with a fast acceleration rate in recent years, reaching a warmer value of 1.80 °C from 2012 to 2020 [41]. In this sense, the high virulence of *Haplosporidium pinnae* on *Pinna nobilis* from 2016, which has practically eradicated all populations in the Balearic Islands, is associated with an increase in water temperatures above 13.5 °C, which favours the development of the endoparasite [42]. In this scenario, in 2015, a fisherman reported for the first time spots on the skin of *X. novacula* caught in the south of the island of Ibiza. The fact that the spots were observed in 2015 does not mean that the parasite was already there before that, although from that moment the number of infected fish and spots has been evident. These spots have been identified as a trematode ectoparasite of the genus *Scaphanocephalus* sp. [26]. Ribosomal DNA analysis is usually used to study genetic differentiation within digenean families [43]. However, publicly available sequences only allow us to attribute our 28rDNA sequences to *Scaphanocephalus* sp. Since the first report, it seems that the infection is spreading its distribution to different areas of the Ibiza Island and also increasing its intensity. In this sense, in a previous study where specimens of *X. novacula* were fished in 2020, in a very close area to those of the present study, the average number of spots was 12.3 per specimen, while two years later the average reached 27.6 [26]. Furthermore, in the specimens with a greater presence of ectoparasites, some black spots appear along with the white spots. These lesions have been described histopathologically as an accumulation of melanin-producing cells called melanophores in the epidermis which encapsulate the parasite cyst [44]. The results of the present work have shown that the presence of the ectoparasite *Scaphanocephalus* sp. in the skin of *X. novacula* induced a progressive increase in the markers of oxidative stress and of the immune response, as the degree of parasitism increased along with a decrease in the body condition index.

The prevalence of ectoparasite infection in the Es Cubells area is very high compared to the majority of previous studies. In this sense, a previous study carried out in Okinawa (Japan), found the *Scaphanocephalus* sp. infection prevalence in different species of the genus *Scarini* (parrot fish), family Labridae, was much lower than those observed in Es Cubells, with a range between 0.1% and 38.5% [16]. Similarly, in another study evaluating the presence of *Scaphanocephalus* sp. in *Siganus argenteus* (streamlined spinefoot), family Siganidae, in the Arabian Gulf, 22.9% of the studied specimens (800 out of 3500) showed black cysts along the body surface [45]. Results similar to those of the present study were

found in *Acanthurus tractus* (ocean surgeonfish) infected with *Scaphanocephalus expansus* on the island of Bonaire (Caribbean Sea) with about 90% of fish affected [14]. To date, there has not been any study carried out in the Mediterranean that evaluates the presence of *Scaphanocephalus* sp. in any species of fish. The few studies that evaluate the presence of parasites of the digenea subclass focus on intestinal parasites and not ectoparasites, and therefore the data are not comparable [46,47]. With these results, it is evident that the high prevalence of *Scaphanocephalus* sp. in the Es Cubells area may pose a risk to the resident population of *X. novacula* in the area, so long-term monitoring studies are necessary. In fact, the effects on *X. novacula* are evident in the reduction of the body composition index, which would be indicative of a worse state of health in fish that suffer a greater degree of infection. In this sense, in a review analysing 16 studies where introduced parasites have been able to infect native hosts and information on relative virulence was available, 85% were more virulent in these new hosts than in the original host [48]. Thus, parasite infection can negatively affect the dynamics and density of fish populations and alter the entire community [49].

When analysing the activity of antioxidant enzymes in the mucus, a progressive increase was observed in the antioxidant enzymes CAT, SOD and GPX with the increased presence of parasites. This fact would indicate that individuals with a higher parasite load have a higher enzyme activity when compared to those with a lower parasite load due to the stress associated with infection. In accordance with these results, in a previous study, an increase in the activities of the antioxidant enzymes CAT, SOD and glutathione reductase was observed in the mucus of *Sparus aurata* (gilthead seabream) before an experimentally induced epithelial injury [50]. Moreover, an induction of the mucosal antioxidant defences has also been reported in fish exposed to pollutants [51]. In another study, an increase in the expression of antioxidant genes such as manganese superoxide dismutase (MnSOD), a natural killer cell enhancing factor, has been observed in the skin of *Labeo rohita* (Indian major carp) infected by the ectoparasitic crustacean *Argulus siamensis* [52]. However, despite this, the increase in the antioxidant defences was not sufficient to prevent an increase in MDA and protein carbonyl levels as an indicator of oxidative damage to lipids and proteins, respectively, in the group of *X. novacula* with a higher degree of parasitism. These results show that if the fish are not able to stop the infection, it can progress and cause oxidative damage, which could end up compromising the immune system of the affected specimen. The importance of the skin barrier was determined in an interesting study, where it was reported that dietary vitamin D administration reduced the skin mucosal levels of MDA and protein carbonyls in *Ctenopharyngodon idella* (grass carp) infected with *Aeromonas hydrophila*, thus enhancing resistance against the infection [53].

The response of the immune system of *X. novacula* to different degrees of infection by *Scaphanocephalus* sp. was evaluated in the mucus and in the spleen by measuring different biomarkers, such as lysozyme, ALP, MPO and the concentration of immunoglobulins. The use of the skin mucus is useful, as it allows monitoring the response of the fish after an early infestation without the sacrifice of the individual [26,54]. Moreover, fish have the ability to change the composition of the mucus and increase its thickness when infected by an ectoparasite [55]. In the present results, all the analysed parameters increased progressively with the infection, showing significant differences between the high infection group and the low infection and non-infected groups. The results highlight the activation of the immune response in *X. novacula* to the presence of the parasite in order to avoid or reduce the progression of the infection. This response agrees with what was previously detected in *X. novacula* parasitised with *Scaphanocephalus* sp., also in areas of Ibiza and Formentera where individuals affected by the parasite had a greater activation of the immune system [26]. Similar results were found in another study where *Pangasianodon hypophthalmus* (iridescent shark catfish) parasitised by dactylogyrid monogeneans of the genus *Thaparocleidus* had higher values of lysozyme, total immunoglobulins and α-2 macroglobulin [56]. Similarly, *Catla catla* (catla) mucosal immunity markers, such as MPO, ALP and anti-protease activities, were significantly higher in fish infected by mono-

geneans parasites, mainly belonging to the genus *Dactylogyrus* [57]. *Epinephelus coioides* (orange-spotted grouper), when parasitised with the protozoan *Cryptocaryon irritans*, presented higher levels of lysozyme and ALP when compared to non-parasitised ones [58]. Very similar results were obtained when the effects of *C. irritans* infection on *Pseudosciaena crocea* (large yellow croaker) were assessed with an increase in the aforementioned immune markers and complement C3 [59]. Similarly, an increase in ALP activity was observed in salmon specimens (*Salmo salar*) as the degree of infestation by the copepod *Lepophtheirus salmonis* increased [35], or in the presence of the bacterium *Aeromonas salmonicida* [60]. Altogether, the results highlight the importance of skin mucus with an increase in immunological protection mechanisms induced by the presence of the parasite.

The spleen has a central role in immune protection in fish, participating in haematopoiesis and in the elimination of exogenous substances and harmful metabolites [21], thus its role in supporting immune responses is fundamental for the overall health and survival of these animals. The present study found an increase in spleen activities of lysozyme and ALP in fish intensely affected by the ectoparasite, while no significant effects were evident if the infection was low, suggesting that it was not enough to activate a systemic response. A previous work reported that grass carp (*Ctenopharygodon idellus*) infected by the monogenean ectoparasite *Dactylogyrus lamellatus* induced significant histological alterations in the spleen of the severely infected fish [61]. Also, the expression of immune-related genes—*MHCII*, *MyD88*, *TLR3* and *IgM*—was significantly up-regulated in the spleen. Likewise, the infection of cultured sea breams (*Sparus aurata*) by the monogenean ectoparasite *Sparicotyle chrisophrii* induced histological changes, with a significant increase in splenic melanomacrophage centres [62].

Finally, it is important to highlight that the correlation analysis shows a direct relationship between the degree of infection, considering the number of epithelial spots observed, and the different parameters of oxidative and immunological stress, and an inverse relationship with the body condition index. These results suggest that the parameters analysed are suitable for monitoring the infection by *Scaphanocephalus* sp. in *X. novacula*. The absence of correlations in the spleen would be mainly due to the fact that a low infection is not enough to induce significant changes, and an epithelial response is initially activated. The lack of correlation between the number of spots and the size of the fish would indicate that there are other factors besides the age and size of the fish that contribute to the degree of infection.

## 5. Conclusions

The present results have highlighted the high prevalence of *X. novacula* specimens affected by the ectoparasite of the genus *Scaphanocephalus* sp. in the Es Cubells area on the island of Ibiza. This high presence of affected fish and the high number of parasites in some individuals suggests that a significant alteration may occur in the community and that the infectious process continues to progress. This infection triggers a progressive activation in the mucus of the antioxidant and immune systems in individuals as the degree of parasitism increases. This activation seems to be effective in the early stages of infection or in cases of moderate infection, while when the degree of infection is very high, the protective mechanisms are not able to prevent an increase in lipid peroxidation, indicative of oxidative stress. Additional studies are needed to identify the specific species of *Scaphanocephalus* that causes the infection, of which mollusc species are the intermediate host that would help predict its potential distribution and continue to monitor how the prevalence on *X. novacula* evolves and if the parasite is extending to other areas.

**Author Contributions:** All authors contributed to the study conception and design; methodology, A.C.-S., J.M.V., S.P., A.B. and A.S.; formal analysis, A.C.-S. and A.G.S.-M.; investigation, A.C.-S., A.G.S.-M., S.T., A.B. and A.S.; writing—original draft preparation, A.C.-S., A.G.S.-M. and A.S.; writing—review and editing, S.P., A.B. and S.T.; project administration, A.S. and S.T.; funding acquisition, S.P., S.T. and A.S. All authors have read and agreed to the published version of the manuscript.

**Funding:** This work has been partially sponsored and promoted by the Comunitat Autònoma de les Illes Balears through the Direcció General de Recerca, Innovació I Transformació Digital and the Conselleria de Economia, Hisenda i Innovació via Plans complementaris del Pla de Recuperació, Transformació i Resiliència (PRTR-C17-I1) and by the European Union- Next Generation UE (BIO/006). Nevertheless, the views and opinions expressed are solely those of the author or authors, and do not necessarily reflect those of the European Union or the European Commission. Neither the European Union nor the European Commission are to be held responsible. This research was partially funded by the Spanish Government, Institute of Health Carlos III (CIBEROBN CB12/03/30038). This work has also been financed by the Biodibal project, within the framework of the Collaboration Agreement between the University of the Balearic Islands and Red Eléctrica de España.

**Institutional Review Board Statement:** The study protocol was approved by the Ethics Committee for Animal Experimentation of the University of the Balearic Islands (Ref. 020/06/AEXP).

**Data Availability Statement:** Researchers wishing to access the data used in this study can make a request to the corresponding author: antoni.sureda@uib.es.

**Conflicts of Interest:** The authors declare no conflict of interest.

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
