# Peer review of "Immune and Oxidative Stress Response of the Fish Xyrichthys novacula Infected with the Trematode Ectoparasite Scaphanocephalus sp. in the Balearic Islands"

_fishes, doi:10.3390/fishes8120600_

Round 1

Reviewer 1 Report

Comments and Suggestions for Authors

I have completed the manuscript review intitule “Immune and oxidative stress response of the fish Xyrichthys novacula induced by a new arrived trematode ectoparasite in the Balearic Islands. I consider it to be a solid piece of work deserving of publication. However, there are some necessary changes that will enhance the manuscript's quality:

1. Explain in the methodology section the procedure used to evaluate the morphology of the found metacercaria.

2. Provide detailed methodology regarding the extraction of parasite DNA, along with the PCR conditions and the specific primers used.

3. In the results section, include a description of the parasite.

4. Include improved photographs of the trematode focusing on details of taxonomic importance.

5. I highly doubt that this species can be considered an ectoparasite; typically, the term refers to the location of the adult form.

Author Response

Reviewer 1

I have completed the manuscript review in title “Immune and oxidative stress response of the fish Xyrichthys novacula induced by a new arrived trematode ectoparasite in the Balearic Islands”. I consider it to be a solid piece of work deserving of publication. However, there are some necessary changes that will enhance the manuscript's quality:

  1. Explain in the methodology section the procedure used to evaluate the morphology of the found metacercaria.

The procedure followed to obtain the image of the metacercariae has been added to the manuscript:

“For excystation of metacercariae from fish skin preserved in ethanol, two syringe needles (30G x ½”) were used and visualized under an Olympus SZH stereomicroscope with a HD camera”.

  1. Provide detailed methodology regarding the extraction of parasite DNA, along with the PCR conditions and the specific primers used.

The required information has been added to the revised version on the manuscript:

“For molecular analysis, a total of 14 metacercaries from different specimens were dissected from epithelial sections containing skin papules using dissecting needles. To confirm whether the different types of spots (black and white) were caused by the same parasite, 8 spots of each type were analysed. DNA extraction was then carried out using a tissue DNA extraction kit (Macherey-Nagel XS) following the instructions provided by the manufacturer. Once the DNA was purified, the 28S rDNA was partially amplified, including the variable expansion regions D1-D3 of the parasites using the polymerase chain reaction (PCR). The used primers were 1500R (5’-GCT ATC CTG AGG GAA ACT TCG-3′) and LSU5 (5′-TAG GTC GAC CCG CTG AAY TTA AGC-3′) (Littlewood et al., 2000; Olson et al., 2003). PCR reactions were performed using the KAPA Taq Ready Mix DNA Polymerase (KapaBiosystems) with the following conditions: a denaturation period of 5 min at 95 â—¦C followed by 40 cycles of 95ºC for 30s, 57ºC for 30s, 72ªC for 1min and a final stage of 72ªC for 10min.

PCR products were separated in 1% agarose and stained with GelRed® Nucleic Acid Gel Stain (Biotium, CA, USA). Then, the obtained amplicons were purified with the mi-PCR purification kit (Metabion International, Germany) and sequenced on a 3130xl automated DNA sequencer (Applied Biosystems, CA, USA). Sequences were finally aligned and edited using the BioEdit 7.1.3.0 software package (Hall, 1999) and compared using the National Center for Biotechnology Information (NCBI) Basic Local Alignment Search Tool (BLAST)”.

Hall, T.A. BioEdit: a user-friendly biological sequence alignment editor and analysis program for Windows 95/98/NT. Nucleic Acids Symp. Ser. 1999, 41, 95-98.

  1. In the results section, include a description of the parasite.

A general description of the parasite and the obtained data from the molecular results was added to the revised version of the manuscript:

“The cysts are oval in shape with a fibrous capsule that surrounds it. On the fish, they appear as whitish spots in the early stages that change to black as the surrounding tissue becomes to accumulate melanin”.

“Adequate amplification was obtained in 8 of the samples analysed (5 coming from white spots and 3 from black spots), resulting in a sequence with 1283 base pairs of the LSU rDNA gene after aligning the sequences of the different PCR amplicons.  The sequences obtained showed an identity between 98.4 and 98.6% with the 4 available sequences of Scaphanocephalus sp. in GenBank (MN160569, MN160569, MT461356, MK680936). GenBank accession numbers for the obtained sequences were OK045681 to OK045688.

Phylogenetic tree of 28rDNA (Figure 4) placed the obtained sequences in a clade with other Scaphanocephalus species (S. expansus, Scaphanocephalus sp. from Acanthurus chirgus and Scaphanocephalus sp. from Siganus argenteus), Cryptocotyle lingua and Euryhelmis costaricensis. All of them are digenean from the family Heterophyidae”.

Figure 4. Maximum likelihood topology of digenean taxa based on 28rDNA partial sequences. Nodes show posterior probabilities (numbers in bold) and bootstrap support values. Novel sequences from this study are in bold and group with Scaphanocephalus expansus, Scaphanocephalus sp and Cryptocotyle lingua, heterophyid digeneans known to cause black spot disease in fishes in the Atlantic and Mediterranean. Philophthalmus gralli was used to root the tree. All GenBank accession numbers are given before the taxon branch labels.

  1. Include improved photographs of the trematode focusing on details of taxonomic importance.

Unfortunately, a taxonomic study has not been carried out based on the morphology of the parasite since the final objective of this work was to demonstrate the effects it produces on fish, especially those in which the degree of infection is very high. It is planned to carry out these morphological studies, so samples have already been sent to specialists in the subject, in order to confirm with certainty, the species as S. expansus.

  1. I highly doubt that this species can be considered an ectoparasite; typically, the term refers to the location of the adult form.

We were also discussing the topic that the reviewer addresses, since the final phase is actually a parasite of the digestive system of seabirds (at least, it is known to affect the osprey, Pandion haliaetus) (Foronda et al, 2009; Koh et al, 2019). However, in the intermediate host such as X. novacula it is an ectoparasite, so in the end we have thought that it is the most correct way, especially based on previous articles that deal with this same parasite in fish. However, in the presentation of the parasite in the introduction we have clarified it by indicating that it is an endoparasite in the final host but that in fish it generates pigmented dermatopathies, presented as focal black circular spots or papules, acting as a cutaneous endoparasite.

 Foronda, P.; Santana-Morales, M.A.; Feliu, C.; Valladares, B. New record of Scaphanocephalus expansus from the Canary Islands (Spain). Helminthologia. 2009, 46, 198-200. Doi:10.2478/s11687-009-0036-5.

 Kohl, Z.F.; Calhoun, D.M.; Elmer, F.; Peachey, R.B.J.; Leslie, K.L.; Tkach, V.; Kinsella, J.M.; Johnson, P.T.J. Black-spot syndrome in Caribbean fishes linked to trematode parasite infection (Scaphanocephalus expansus). Coral Reefs 2019, 38. Doi:10.1007/s00338-019-01819

Reviewer 2 Report

Comments and Suggestions for Authors

Review of the manuscript: ' Immune and oxidative stress response of the fish Xyrichthys novacula induced by a new arrived trematode ectoparasite in the Balearic Islands ' by Amanda Cohen-Sánchez and colleagues. I have a few questions and comments about the experimental design and sampling section of this paper before being further concerned. More work is needed to substantiate the conclusions in your manuscript.

 The primary issue with the paper lies in the experimental design, which can be deemed as lacking in reasonability. In this paper, the wild-caught fish were used in this experiment and these fish species may harbour a variety of endoparasite, such as intestinal parasites. We must clarify the negative impacts on wild experimental fish in our experimental design will be derived only from an ectoparasite. Thus, how can the authors exclude the hypothesis that potential biases caused by endoparasite, are the cause for these differences? Moreover, it seems from the methods that the control group and the experimental group of fish were sourced from distinct aquatic environments. How do the authors comment on that or how do they exclude the impact of the different backgrounds on experiment animals?

 Line 38. “… smooth activity”? I don't understand what that means.

 Line 51-55. Please include citation.

 Line 96-98. Please include citation.

 Line 101-103. “…notable increase in the presence of Scaphanocephalus”. if the statement made here is based on the previous study, then please include citation.

 Line 127. where are these mucus samples coming from specifically?

 Line 132. I don't understand the reason of the infection severity chosen. How to define “low infection”? What basis?

 Description of 2.1 Samplingwas too redundant and authors should give a topic sentence for each step.

 In the sampling section, add a reference to the MSS222 dose. Add the repetition number of biochemical indicators.

 Line 181-182. Each experimental group only contains ten fish. The number of samples is too small to be statistically significant. This may not be representative.

 Line 257-258. in a previous study? Please include citation.

 When looking at the results referring to oxidative stress I am not convinced. Considering MDA as a biomarker of oxidative damage I am not convinced for two reasons: (1) there are no complementary oxidative stress biomarkers like DCF-DA and NRF2 expression, for example. (2) The MDA can be generated by onsuccessive hydroperoxide formations and b-cleavage of the fatty acid chain to give a hydroperoxyaldehyde. Thus, it is possible that instead of high levels of MDA being related to oxidative stress, they are related to an increase in b-oxidation.

 The "p" values in the manuscripts are not italicized.

 The discussion section has contained more general information. Discussion of the findings was considered insufficient.

Author Response

Reviewer 2

Review of the manuscript: ' Immune and oxidative stress response of the fish Xyrichthys novacula induced by a new arrived trematode ectoparasite in the Balearic Islands ' by Amanda Cohen-Sánchez and colleagues. I have a few questions and comments about the experimental design and sampling section of this paper before being further concerned. More work is needed to substantiate the conclusions in your manuscript.

The primary issue with the paper lies in the experimental design, which can be deemed as lacking in reasonability. In this paper, the wild-caught fish were used in this experiment and these fish species may harbour a variety of endoparasite, such as intestinal parasites. We must clarify the negative impacts on wild experimental fish in our experimental design will be derived only from an ectoparasite. Thus, how can the authors exclude the hypothesis that potential biases caused by endoparasite, are the cause for these differences?

To avoid possible interference in the results due to the presence of endoparasites, the internal cavity of each fish was carefully checked and only those that did not have any evidence of endoparasites were chosen for future analyses. This information has been clarified in the revised version of the manuscript.

“Additionally, the intestinal cavities of all initially selected specimens were inspected to detect the possible presence of endoparasites that could interfere with the study. Endo-parasites were observed in three fish from Es Cubells that were directly discarded for the study”.

Moreover, it seems from the methods that the control group and the experimental group of fish were sourced from distinct aquatic environments. How do the authors comment on that or how do they exclude the impact of the different backgrounds on experiment animals?

A control area was selected where there was no evidence of the parasite, since in the affected area of Es Cubells practically all the specimens appeared with some stain on their epithelium. From Es Cubells, exploratory sampling was carried out to determine how far away infected fish were caught and to establish a control area without evidence of the parasite. Thus, a nearby area was selected where the fish could be obtained at the same depth (18-20 meters) and on the same sampling days to try to minimize potential interference due to other factors. In the methodology section, new information has been incorporated to try to clarify it.

Line 38. “… smooth activity”? I don't understand what that means.

This is a typographic error, and refers to lysozyme activity, this has been corrected in the revised version of the manuscript. 

Line 51-55. Please include citation.

According to the reviewer comment, a reference has been added:

Turbelin, A. J., Malamud, B. D., & Francis, R. A. (2017). Mapping the global state of invasive alien species: patterns of invasion and policy responses. Global Ecology and Biogeography, 26(1), 78-92.

Line 96-98. Please include citation.

No reference can be incorporated into this sentence since these are observations carried out by members of the research group. In the revised version, we have indicated that these are personal observations. 

Line 101-103. “…notable increase in the presence of Scaphanocephalus”. if the statement made here is based on the previous study, then please include citation.

The required citation has been added to the text:

Cohen-Sánchez, A.; Valencia, J.M.; Box, A.; Solomando, A.; Tejada, S.; Pinya, S.; Catanese, G.; Sureda, A. Black Spot Disease Related to a Trematode Ectoparasite Causes Oxidative Stress in Xyrichtys Novacula. J. Exp. Mar. Bio. Ecol. 2023, 560, 151854.

Line 127. where are these mucus samples coming from specifically?

This information has been added to the revised version “Due to the small size of the fish, the mucus has been obtained from the dorsal area of both sides of the specimens”. We also specified that mucus was obtained from the skin using a spatula.

Line 132. I don't understand the reason of the infection severity chosen. How to define “low infection”? What basis?

The two groups have been established according to the median of the spots of the infected fish since it is the value of the variable that allows us to have two similar groups with 50% of fish above and 50% below this value. This information has been added to the revised version. 

Description of “2.1 Sampling” was too redundant and authors should give a topic sentence for each step.

This was changed to sample recollection.  

In the sampling section, add a reference to the MSS222 dose. Add the repetition number of biochemical indicators.

The reference has been added:

Capó, X., Alomar, C., Compa, M., Sole, M., Sanahuja, I., Soliz Rojas, D. L., González, G. P., Garcinuño Martínez, R. M., & Deudero, S. (2022). Quantification of differential tissue biomarker responses to microplastic ingestion and plasticizer bioaccumulation in aquaculture reared sea bream Sparus aurata. Environmental research, 211, 113063. https://doi.org/10.1016/j.envres.2022.113063

The repetition number of indicators has been added. All biochemical analysis were performed in duplicate. 

Line 181-182. Each experimental group only contains ten fish. The number of samples is too small to be statistically significant. This may not be representative.

Nowadays, the laws in our country are very specific with the number of animals used (under or below the necessary). So, one of the objectives is to reduce the number of animals used in research and it should be authorized by the authority; in this sense, we have experience and we performed previous works with similar animals in which the sample size was approximately the same as the used in the current work (Box et al., 2020a and 2020b). Moreover, the statistical analysis showed that the data was consistent. With experimental animals a method of calculation is a crude method based on law of diminishing return. This method is called “resource equation” method. It is used when it is not possible to assume about effect size, to get an idea about standard deviation as no previous findings are available or when multiple endpoints are measured or complex statistical procedure is used for analysis. This method can also be used in some exploratory studies where testing of hypothesis is not the primary aim, but researcher is interested only in finding any level of difference between groups. According to this method a value “E” is measured, which is nothing but the degree of freedom of analysis of variance (ANOVA). The value of E should lie between 10 and 20. If E is less than 10 then adding more animals will increase the chance of getting more significant result, but if it is more than 20 then adding more animals will not increase the chance of getting significant results. Though, this method is based on ANOVA, it is applicable to all animal experiments. Any sample size, which keeps E between 10 and 20 should be considered as an adequate. E can be measured by following formula: E = Total number of animals − Total number of groups.

We wanted to observe the effect of the infection in the fish, so we made 3 groups (uninfected, low infection and high infection) with 10 animals each. In this case E was: E = (10 × 3) – 4 = 16 => It is between 10 and 20, so that we considered it adequate. Then, as this kind of fish are protected by law and there is a concrete period in which they can be caught, we assume a conservative number of animals. For more information about method see (Charan & Kantharia, 2013).

Box, A., Capó, X., Tejada, S., Sureda, A., Mejías, L., & Valencia, J. M. (2020a). Perkinsus mediterraneus infection induces oxidative stress in the mollusc Mimachlamys varia. Journal of fish diseases, 43(1), 1-7.

Box, A., Capó, X., Tejada, S., Catanese, G., Grau, A., Deudero, S., Sureda, A., & Valencia, J. M. (2020b). Reduced Antioxidant Response of the Fan Mussel Pinna nobilis Related to the Presence of Haplosporidium pinnae. Pathogens (Basel, Switzerland), 9(11), 932.

Charan J, Kantharia ND. How to calculate sample size in animal studies? J Pharmacol Pharmacotherap 2013;4(4):303-306. doi: 10.4103/0976-500X.119726.

Line 257-258. “in a previous study”? Please include citation.

The required citation has been added:

Cohen-Sánchez, A.; Valencia, J.M.; Box, A.; Solomando, A.; Tejada, S.; Pinya, S.; Catanese, G.; Sureda, A. Black Spot Disease Related to a Trematode Ectoparasite Causes Oxidative Stress in Xyrichtys Novacula. J. Exp. Mar. Bio. Ecol. 2023, 560, 151854.

When looking at the results referring to oxidative stress I am not convinced. Considering MDA as a biomarker of oxidative damage I am not convinced for two reasons: (1) there are no complementary oxidative stress biomarkers like DCF-DA and NRF2 expression, for example. (2) The MDA can be generated by on successive hydroperoxide formations and b-cleavage of the fatty acid chain to give a hydroperoxyaldehyde. Thus, it is possible that instead of high levels of MDA being related to oxidative stress, they are related to an increase in b-oxidation.

Since mucus is an exudate generated by fish as a protective mechanism, it is not expected that there may be derivatives of β-oxidation, which is why we think that MDA is an indicator of oxidative damage. In this sense, different manuscripts that have determined MDA in mucus relate it to oxidative damage to lipids. Some manuscripts that indicate this are:

Yu, Z., Hao, Q., Liu, S. B., Zhang, Q. S., Chen, X. Y., Li, S. H., ... & Zhou, Z. G. (2023). The positive effects of postbiotic (SWF concentration®) supplemented diet on skin mucus, liver, gut health, the structure and function of gut microbiota of common carp (Cyprinus carpio) fed with high-fat diet. Fish & shellfish immunology, 135, 108681. 

Singh, J., Srivastava, A., Nigam, A. K., Kumari, U., Mittal, S., & Mittal, A. K. (2023). Alterations in certain immunological parameters in the skin mucus of the carp, Cirrhinus mrigala, infected with the bacteria, Edwardsiella tarda. Fish Physiology and Biochemistry, 1-18.

To improve the section related to oxidative damage, the analysis of cabonyl groups has been carried out as an indicator of oxidative stress in proteins. The results obtained have been incorporated into the revised version of the manuscript and show a response pattern to the MDA. Furthermore, and also in accordance with the comment that follows, the discussion on this matter has been expanded.

The "p" values in the manuscripts are not italicized.

This was corrected. 

The discussion section has contained more general information. Discussion of the findings was considered insufficient.

The entire discussion section has been revised trying to go deeper and be more specific according to the reviewer's comment.

Reviewer 3 Report

Comments and Suggestions for Authors

Comments on the Quality of English Language

Some improvements to the quality of English will need to occur.  I have made some suggested changes in my review.

Author Response

Reviewer 3

Review of “Immune and oxidative stress response of the fish Xyrichthys novacula induced by a new arrived trematode ectoparasite in the Balearic Islands”

This manuscript compares immunological markers in the skin mucus and spleen of pearly razorfish at a site where Scaphanocephalus was (Es Cubells) and was not (Sa Mola) present. In fish with high infections (> 15 cysts), there was a decrease in condition factor and in skin mucus, SOD, CAT, and GPX significantly increased. My main concerns are addressed below. My top concern is that size was not correlated with parasite abundance and with the other markers assessed in the fish (there has been lots of studies that show as fish get larger, they accumulate more parasites). Fish size should also be compared between sites. My other major concern is that the methods are not detailed enough (especially when it comes to explaining sample size). I do not feel like 10 fish at a site at one time point in one year is enough to convey statistical relevancy, but if you can explain your reasoning and whether statistics were used to determine sample size, that will suffice. Overall, I believe if major revisions are completed, the manuscript should be suitable for publication.

 Questions/Comments

Methods

2.1 Sampling

  • Where any other types of parasites present in any other tissues?

The internal cavity of the fish was checked and, if other types of parasites were found, these fish were discarded.

“Additionally, the intestinal cavities of all initially selected specimens were inspected to detect the possible presence of endoparasites that could interfere with the study. Endo-parasites were observed in three fish from Es Cubells that were directly discarded for the study”.

  • Why didn't you include histology to see what cellular changes were associated with this ectoparasite? Many times, there is very minimal host response to these parasites (generally mild inflammation and melanophore accumulation).

We agree with the reviewer that histological analysis would be really interesting; however, we have not carried it out as we are not experts on the subject. At this time the study continues as the evolution of the infection will be monitored over the years, in addition to histological analyzes in fish and taxonomic analyzes in the parasite in collaboration with other researchers who are experts in the subject.

  • Your numbers don’t make sense. You looked at 10 females from Sa Mola and 20 females from Es Cubells, that is not a similar sample size. Also, 77-10-6=61. What happened to the 41 fish that were caught but not used in this study and why weren’t they used? This is very confusing as worded, please provide clarification.

After reading the segment we agree with the reviewer that the form of expression was not the most appropriate. We have revised the text to make the specimen selection process clearer. The fish that were not used for the subsequent biochemical analysis were returned to the sea, but were used for the infection prevalence study.

  • Why were the males discarded? Wouldn't it have been worthwhile to do a sex comparison?

The males were not used in the study, since the number of captures was very low, with females predominating in the study areas. Thus, and to use fish of the same size and reduce bias due to size and potential exposure time to the parasite since they are protogynous sequential hermaphrodites, it was decided to work only with females. The low presence of males in the fisheries has been indicated in the revised text.

  • Approximately how much mucus was collected from each fish?

About 0.75 mL of mucus was collected, filling approximately half of a 1.5 mL tube. The obtained volume has been indicated in the text.

  • At what concentration was the MS-222 used? Was it buffered?

MS-222 was used in a concentration of 1g/10L marine water in accordance with previous study that has been cited in the text:

Capó, X., Alomar, C., Compa, M., Sole, M., Sanahuja, I., Soliz Rojas, D. L., González, G. P., Garcinuño Martínez, R. M., & Deudero, S. (2022). Quantification of differential tissue biomarker responses to microplastic ingestion and plasticizer bioaccumulation in aquaculture reared sea bream Sparus aurata. Environmental research, 211, 113063. https://doi.org/10.1016/j.envres.2022.113063

  • In your paper "Black spot disease related to trematode ectoparasite causes oxidative stress in Xyrichtys novacula," low infection is characterized as 0-1 parasites and high infection is >/= 7 parasites. Why did you change that in this paper? Additionally, there is a paper that came out before yours in 2019 that provides a range of infectivity, why didn't you use that to stage Scaphanocephalus infections? "Black spot syndrome in Caribbean fishes linked to trematode parasite infection (Scaphanocephalus expansus)." You need to provide justification for why you chose these stages as opposed to the ones previously published. Essentially, coming up with new staging makes it so the findings from these previous publications are not relatable to this study.

The two groups have been established according to the median of the spots of the infected fish since it is the value of the variable that allows us to have two similar groups with 50% of fish above and 50% below this value. We used this selection criterion to have the three experimental groups balanced according to the samples we had obtained. Furthermore, since they are different species that belong to different families and the average sizes of both species are different, using the same criterion may not be appropriate since in each case the conditions in which the fish are found are different. This information has been added to the revised version.

  • Please brighten Figure 2B and 2C.

Thank you for the suggestion, both images have been brightening.

  • The grubs in 2B and 2C appear to be white grubs while in 2C there also appears to be black spots. Did you sequence the white spots and the black spots to confirm that they are not two different types of parasites? Given the white spots are so much larger, could the black spots be another type of monogenean trematode (such as a Posthodiplostomum or Diplostomum which also cause black spot disease)?

Samples from different types of spots have been sequenced and the results are the same, pointing to S. expansus. Furthermore, spots have subsequently been analyzed in other wrasse species that have begun to develop spots, such as Thalassoma pavo and Coris julis, and the results are similar. In this sense, the change in color seems to be more related to the time or progression of infection since dark colors are not observed in those individuals that have only few spots. Additional information about molecular identification has been added to the text to better explain this issue.

  • Figure 2C: please include arrows pointing to some of the white spots. This is a heavy infection, so much so that the reader could overlook that those are all individual parasites.

Three arrows have been included to point some white spots as it has been suggested.

  • Histologically, the black pigment associated with the parasite is melanin (melanophores) which encapsulates the parasite cyst, it is not necrosis, please revise.

We thank the reviewer for clarifying the black pigment associated with the parasite. Thus, in the revised version of the manuscript it has been corrected indicating in figure 2 that the coloration is due to melanin and not necrosis. In addition, new information has been added to the discussion section:

“Furthermore, in the specimens with a greater presence of ectoparasites, some black spots appear along with the white spots. These lesions have been described histopathologically as an accumulation of melanin-producing cells called melanophores in the epidermis which encapsulates the parasite cyst”.

The following reference has been incorporated:

Noguera, P. A., Feist, S. W., Bateman, K. S., Lang, T., Grütjen, F., & Bruno, D. W. (2013). Hyperpigmentation in North Sea dab Limanda limanda. II. Macroscopic and microscopic characteristics and pathogen screening. Diseases of aquatic organisms, 103(1), 25–34. https://doi.org/10.3354/dao02553

  • Please confirm whether you examined inside the operculum for black spots, they can often be found there too.

Inside the operculum has been examined for black spots but in none of the fish was evidence of spots. This information has been incorporated into the new version of the manuscript.

  • How was the DNA extracted and at what concentration? How many metacercaria were used?

The information about the DNA extraction and molecular analysis has been expanded and improved in the revised version of the manuscript.

  • Please give the reader some information on why you are testing for all of these markers (lines 149-166)? What significance do they have to fish health and parasitism?

Some additional information has been added to the text to better focus the makers analysed:

“The presence of parasites can cause physiological and behavioural alterations in hosts, favouring new infections or their predation, endangering their survival. To cope with ectoparasites, fish have developed an external mucosal layer that acts as both a physical and biochemical barrier. Among the components of mucus are immune elements such as immunoglobulins and lysozyme and antioxidant enzymes to reduce oxidative damage derived from the infectious process. In this sense, immunological parameters and markers of the oxidative state can provide information on the evolution of the infection and the host's response capacity”.

  • Also, why was the spleen chosen? What significance does it have for fish immunity? I know why, but another reader may not.

Information about the spleen and its relevance in immunity has been added to the text for a better understanding.

The spleen is a lymphoid organ in fish involved in immune reactivity and blood cell formation. Also, it has been Spleen size has also been suggested that the spleen responses can reflect adaptation to the level of parasite load in the host population (Morand & Poulin, 2000).

Morand, S., & Harvey, P. H. (2000). Mammalian metabolism, longevity and parasite species richness. Proceedings of the Royal Society of London. Series B: Biological Sciences, 267(1456), 1999-2003.

2.3 Statistical Analysis

  • Please look at the correlations between size with infection severity and all other markers. Oftentimes, parasite loads increase with size/age.

In the new version, the correlation between the number of spots and the size of the fish has been analyzed. No correlation has been observed. The information has been incorporated into the text.

  • Please describe the methods for the bivariate correlation analyses.

The methods for bivariate correlations were added to the statistical section.

Results

3.1 Fish characteristics

  • What is the range (for both length and weight)?

The range for length and weight has been added to the results.

  • Please provide the sizes for fish at each site and a statistical comparison.

The size and the statistical analysis of the fish at each site has been incorporated to the revised version.

  • Where are your sequencing results? How long of a sequence were you able to obtain? What species was it most similar to in GenBank? I don’t see them anywhere in the Results section.

All this information about the sequencing results has been added to the revised version of the manuscript. Also, the methodology has been expanded and better explained.

3.2 Oxidative stress parameters

  • Should the commas in figure 4C be decimal points?

This has been a mistake since our version of Excel by default incorporates commas instead of points. The graph has been corrected in the new version.

Discussion

  • Do you have any evidence that globalization and/or climate change is occurring in this area? Can you provide some references (specific to this area) that support that? Do you have any historical data that will show that temperatures are warming in this region? Do you have any data that shows land use is changing in the impacted site compared to the control site? Any evidence of contamination differences between the two sites?

In the revised version of the manuscript we have added additional information to support the climate change in Balearic Islands:

“At surface level, the marine heat waves indices have considerably increased over the last four decades from 1982 to 2020 in Balearic Islands, with a fast acceleration rate in recent years reaching a warmer value of 1.80ºC from 2012 to 2020. In this sense, the high virulence of Haplosporidium pinnae on Pinna nobilis from 2016, which has practically eradicated all populations in the Balearic Islands, is associated with an increase in water temperatures above 13.5ºC, which favours the development of the endoparasite”.

The following references were also added:

Pastor, F., Valiente, J. A., & Khodayar, S. (2020). A warming Mediterranean: 38 years of increasing sea surface temperature. Remote sensing, 12(17), 2687.

Juza, M., Fernández-Mora, À., & Tintoré, J. (2022). Sub-Regional marine heat waves in the Mediterranean Sea from observations: Long-term surface changes, Sub-surface and coastal responses. Frontiers in Marine Science, 9, 785771.

Cabanellas-Reboredo, M., Vázquez-Luis, M., Mourre, B. et al. Tracking a mass mortality outbreak of pen shell Pinna nobilis populations: A collaborative effort of scientists and citizens. Sci Rep 9, 13355 (2019). https://doi.org/10.1038/s41598-019-49808-4.

  • Just because Scaphanocephalus was reported for the first time in 2015 does not mean it wasn't there before and no one reported it.

This fact has been added to the text is accordance with the reviewer suggestion.

  • If you sequenced this parasite, why weren't you able to get it to the species level? If it did not match anything in GenBank, why haven’t you reported it as a new species?

Information about the parasite has been incorporated into the revised version of the manuscript. The obtained sequences showed an identity between 98.4 and 98.6% with the 4 available sequences of Scaphanocephalus sp. in GenBank. However, to accurately confirm the species, a morphological analysis of the parasite should be carried out.

  • You finally discuss why you chose the spleen in this study, but what about the main immune-regulating organ, the anterior kidney?

This is true that the kidney is a valuable organ with key functions related to immune-endocrine interactions. We agree that the kidney would also be a very interesting organ to evaluate the systemic effects of Scaphanocephalus infection. However, in a previous work the reviewers recommended to work with the spleen, so we thought it appropriate to analyze this organ. In this sense, as the study will continue for several years, we will begin to collect kidney samples in the subsequent sampling season to analyze their response.

General Edits

Lines 1-3: Change title to “Immune and oxidative stress response of the fish Xyrichthys novacula infected with the trematode ectoparasite Scaphanocephalus sp. in the Balearic Islands”

This was corrected.

Line 28: Change to “in the skin mucus…”

This was corrected.

Line 35: Remove “as”

This was corrected.

Line 39: Change to “…fish with a greater severity of infection was observed.”

This was corrected.

Line 39-41: Change to “In conclusion, as the severity of Scaphanocephalus sp. infection increased, it induced an immune and oxidative stress response in skin mucus and lead to a decrease in overall body condition.”

This was corrected.

Line 41-42: Change to “The potential health effects that the ectoparasite may have on X. novacula populations will require follow-up studies.”

This was corrected.

Line 59: Remove “Particularly” and capitalize “The” at the beginning of the next sentence.

This was corrected.

Line 61: In addition to vessels, what about transmission by animals (such as birds) and by fish movement?

This possibilities have been added to the revised text. Also, the following reference has been added:

Fèvre EM, Bronsvoort BM, Hamilton KA, Cleaveland S. Animal movements and the spread of infectious diseases. Trends Microbiol. 2006 Mar;14(3):125-31. doi: 10.1016/j.tim.2006.01.004.

Lines 79-81: Change to “More than 30 species of ectoparasites have been found to cause blackspot disease, one of which is the genus Scaphanocephalus, a group of ectoparasitic trematodes of the subclass digenea.”

This was corrected.

Lines 81-82: Change to “Digenean trematodes have a complex life cycle that involves several hosts and developmental stages.”

This was corrected.

Line 83-84: Remove “probably” and change “first intermediate” to “primary” and “second” to “secondary host”

This was corrected.

Line 85: Change “from” to “in”

This has been corrected.

Line 86: remove “to” before “marine fish” and remove “particularly wrasses”

This has been corrected.

Line 87-88: Change to “…[12], and even in freshwater fish such as Esox lucius…”

This has been corrected.

Line 88-89: Change to “Up to now, only three species of this genus have been described, including…”

This has been corrected.

Line 92: Change to “…which can grow…”

This has been corrected.

Line 93-94: Change to “…Balearic Islands, both for human consumption and recreational fishing…”

This has been corrected.

Line 99: Change “how” to “that”

This has been corrected.

Line 108: Change to “Pityusic Islands, Spain…”

This has been corrected.

Line 112: Remove “it is” and change to “represents”

This has been corrected.

Line 119-120: Change to “Fish were caught by hook and line with worms as bait.”

This has been corrected.

Line 122: Change “present” to “have” 4

This has been corrected.

Line 127-128: Change to “…with a spatula while avoiding the scales…”

This has been corrected.

Line 128: Change to “…weighed and measured…”

This has been corrected.

Line 137: Change to “the mucus samples were placed…”

This has been corrected.

Line 165: Change “using” to “with”

This has been corrected.

Line 172: Change “show” to “showed”

This has been corrected.

Line 200: Change “no infected” to “non-infected”

This has been corrected.

Line 257: Change to “in an area very close to those of the present study, the average number of spots was 12.3 per specimen, while two years later the average reached 27.6.”

This has been corrected.

Line 273-274: Change to “digenea subclass, including treating intestinal parasites..”

This line has been changed to “The few studies that evaluate the presence of parasites of the digenea subclass, focus on intestinal parasites and not ectoparasites and therefore the data are not comparable”.

Line 286: Change to “…was observed in CAT, SOD, and GPX when parasites increased.”

This has been corrected.

Line 288: Change to “…with a lower parasite load due to the stress associated with infection.”

This has been corrected.

Line 293: Change to “…has also been…”

This has been corrected.

Line 294: Write out what “MOD” stands for

This has been corrected.

Line 299-300: I think you can only go so far as to say it can end up compromising the immune system of affected individuals.

The sentence has been modified in accordance with the reviewer comment.

Line 315: Remove “it was also observed similar results”

This has been corrected.

Line 332: Change “evidenced” to “found”

This has been corrected.

Some improvements to the quality of English will need to occur.  I have made some suggested changes in my review.

The English was revised by a native speaker.

Round 2

Reviewer 1 Report

Comments and Suggestions for Authors

After receiving the corrections from the authors, I am satisfied and recommend publishing this work. However, I do recommend a review of the English.

Author Response

Reviewer 1

After receiving the corrections from the authors, I am satisfied and recommend publishing this work. However, I do recommend a review of the English.

The authors thank the reviewer for the time dedicated to reviewing the manuscript. In agreement with the comment the manuscript has been revised with the help of a Native American postdoctoral researcher to correct the language.

Reviewer 2 Report

Comments and Suggestions for Authors

Authors have successfully addressed all concerns I raised.

Author Response

Reviewer 2

Authors have successfully addressed all concerns I raised.

The authors thank the reviewer for the time dedicated to reviewing the manuscript, which has helped us improve it substantially.

Reviewer 3 Report

Comments and Suggestions for Authors

Overall I feel that the paper is much improved; however, I still have a few suggestions prior to its approval for publication.

Line 62: Change "species" to "animals

Line 102-103: Change to "In fact, it has also been suggested that splenic responses can reflect adaptation to the level of parasite load in the host [22]."

Lines 135-147:

So, were the intestinal cavities of 61 fish examined or was it only the 20 fish that you selected to conduct all of your analyses on?  Also, did you examine their gills as well and inside the stomach/gut for other parasites?  

I think you should include your explanation of why males weren't included within the text.

I think it would have been a really good opportunity to run all of your analyses on the fish from Es Cubells that did not have parasites, that would have helped you to determine whether the differences you observed between the control site and Es Cubells was due to the presence of the parasite or whether it was actually due to site differences (i.e. compare your results from the fish with no parasites at Es Cubells to the fish at the control site). Please explain why you chose not to do this.

After 3 fish with intestinal parasites were discarded, you had a total of 58 fish.  Of these 58 fish, 20 fish of similar size to those from the control site were selected with the presence of spots in order to do further immune analyses, etc..  Did you count the number of parasites on the 38 fish that were returned to the sea?

Line 229: Change "have been" to "were"

Line 240: Change "becomes" to "starts"

Phylogenetic tree: Why does it say "Eivissa Island" and not "Es Cubells?"

Comments on the Quality of English Language

Minor revisions on the quality of English language still need to occur

Author Response

Reviewer 3

Overall I feel that the paper is much improved; however, I still have a few suggestions prior to its approval for publication.

Line 62: Change "species" to "animals

According to the comment the term has been changed.

Line 102-103: Change to "In fact, it has also been suggested that splenic responses can reflect adaptation to the level of parasite load in the host [22]."

The sentence has been changed according to the suggestion.

Lines 135-147:

So, were the intestinal cavities of 61 fish examined or was it only the 20 fish that you selected to conduct all of your analyses on?  Also, did you examine their gills as well and inside the stomach/gut for other parasites?  

Because the fish that were not used for biochemical analyzes had to be returned to the sea, the first 20 specimens were analyzed, but when endoparasites were found in three of them, they were replaced by three new fish. This procedure followed has been incorporated in the revised version of the manuscript.

I think you should include your explanation of why males weren't included within the text.

The explanation has been included in the revised manuscript.

I think it would have been a really good opportunity to run all of your analyses on the fish from Es Cubells that did not have parasites, that would have helped you to determine whether the differences you observed between the control site and Es Cubells was due to the presence of the parasite or whether it was actually due to site differences (i.e. compare your results from the fish with no parasites at Es Cubells to the fish at the control site). Please explain why you chose not to do this.

In the end it was decided to catch the fish without parasites in an area where there were still no signs of their presence. Using the unspotted fish from the Es Cubells area did not give us the certainty that they had not been in contact with the parasite or that they were in an initial phase of infection. We explained this explanation in the revised version.

Next year, we are going to continue with the monitoring when the ban opens in September, and we are going to monitor practically the entire Island, and we are going to repeat some determinations and in more areas, and we will include controls of the affected area and the affected area to increase knowledge about the evolution of the parasite.

After 3 fish with intestinal parasites were discarded, you had a total of 58 fish.  Of these 58 fish, 20 fish of similar size to those from the control site were selected with the presence of spots in order to do further immune analyses, etc..  Did you count the number of parasites on the 38 fish that were returned to the sea?

Ectoparasites were counted in all fish caught to have a more approximate idea of the degree of infection, although only a subsample was used for subsequent analyses.

Line 229: Change "have been" to "were"

It has been corrected.

Line 240: Change "becomes" to "starts"

The term has been changed.

Phylogenetic tree: Why does it say "Eivissa Island" and not "Es Cubells?"

The reviewer is correct with his assessment, so the phylogenetic tree has been modified to indicate that the specimens are from the Es Cubells area.

Minor revisions on the quality of English language still need to occur

The manuscript has been revised with the help of an American postdoctoral researcher to improve grammar.